# Clinicopathological Assessment of Cancer/Testis Antigens NY-ESO-1 and MAGE-A4 in Highly Aggressive Soft Tissue Sarcomas

**DOI:** 10.3390/diagnostics12030733

**Published:** 2022-03-17

**Authors:** Kazuhiko Hashimoto, Shunji Nishimura, Tomohiko Ito, Masao Akagi

**Affiliations:** Department of Orthopedic Surgery, Kindai University Hospital, Osakasayama 589-8511, Japan; shunnisi@med.kindai.ac.jp (S.N.); tomo0251118zooo@gmail.com (T.I.); makagi@med.kindai.ac.jp (M.A.)

**Keywords:** soft tissue sarcoma, antigens, NY-ESO-1, MAGE-A4

## Abstract

We aimed to investigate the clinical significance of the expression of NY-ESO-1 and MAGE-A4 in soft tissue sarcoma (STS). Immunostaining for NY-ESO-1, MAGE-A4, and Ki67 was performed using pathological specimens harvested from 10 undifferentiated pleomorphic sarcoma (UPS), nine myxofibrosarcoma (MFS), and three malignant peripheral nerve sheath tumor (MPNST) patients treated at our hospital. We examined the correlation of NY-ESO-1 and MAGE-A4 expression levels with tumor size, histological grade, and SUVmax values. Positive cell rates of various markers were also compared between patients in remission and those who were not in remission. The rates of cases positive for NY-ESO, MAGE-A4, and Ki67 were 50%, 63.6%, and 90.9%, respectively. The average rates of cells positive for NY-ESO, MAGE-A4, and Ki67 in all STS types were 18.2%, 39.4%, and 16.8%, respectively. A positive correlation was observed between rates of cells positive for NY-ESO-1 and MAGE-A4 and between NY-ESO-1 and MAGE-A4 expression levels and clinical features. There was no significant difference in the positive cell rate of NY-ESO-1 or MAGE-A4 between remission and non-remission cases. Our results suggest that NY-ESO-1 and MAGE-A4 expression may be useful for the diagnosis and prognostication of UPS, MFS, and MPNST.

## 1. Introduction

Soft tissue sarcoma (STS) is a rare disease with a wide variety of histological types and it accounts for approximately 1% of adult malignancies [1,2]. The curative treatment for STS is wide resection, which is combined with chemotherapy and radiotherapy if necessary [3,4]. In general, the drugs used for chemotherapy include doxorubicin, ifosfamide, gemcitabine, and docetaxel [4,5]. Although pazopanib, trabectedin, and eribulin have also been approved, their efficacy is limited [6,7,8]. The median overall survival in patients with unresectable advanced STS remains poor at approximately 1–2 years [9]. Therefore, there is a need to develop novel STS treatments.

Cancer/testis antigens (CTAs) belong to a group of tumor antigens whose normal expression is restricted to male germ cells in the testes and are not found in adult somatic tissues [10]. In addition to their tissue-specific expression profile, among the common features of CTAs are their existence as a multigene family, frequent mapping to the X chromosome, induction of expression by hypomethylation and histone acetylation, immunogenicity in cancer patients, heterogeneous protein expression in various tumor types, and likely correlation with tumor progression [10]. Notably, the CTA epitope is recognized by autologous T lymphocytes that target cancer cells. Therefore, in the last decade, CTAs have emerged as therapeutic targets in treating malignant diseases [11].

New York esophageal squamous cell carcinoma-1 (NY-ESO-1) is an immunogenic CTA associated with innate and vaccine-induced immunity that may lead to clinical cancer responses [12]. Recently, the abnormal expression of NY-ESO-1 has been reported in various neoplasms, including hepatocellular carcinoma, esophageal cancer, melanoma, and non-small cell lung cancer [13].

The melanoma antigen gene (MAGE) protein family comprises a large, highly conserved group of proteins with a common MAGE homology domain [14]. MAGE-A is a CTA and Type I MAGE, which in humans includes members of the MAGE-A, MAGE-B, and MAGE-C subfamilies clustered on the X chromosome [14]. Although the expression of many MAGE proteins is restricted to reproductive tissues, similar to NY-ESO, they have been reported to be aberrantly expressed in various cancers. MAGE-A4 is widely expressed in many tumor types, including esophageal (60%), ovarian (47%), lung (19–35%), colorectal (22%), and breast (13%) cancers [14,15].

Recently, the expression of NY-ESO-1 and MAGE-A4 in STS has been reported, mainly in synovial sarcoma and myxoid liposarcoma [16,17,18,19]. The expression of NY-ESO-1 and MAGE-A4 has also been reported in pediatric STS, including neuroblastoma, Ewing’s sarcoma, and rhabdomyosarcoma [20]. Furthermore, the expression of these two CTAs was shown in the sarcoma of the uterus [21]. It has also been reported to occur in highly aggressive sarcomas such as undifferentiated pleomorphic sarcoma (UPS) [22], myxofibrosarcoma (MFS) [23], and malignant peripheral sheath tumor (MPNST) [24]; however, the number of reports and cases is very small, and the clinical significance of NY-ESO-1 and MAGE-A4 expression has not been investigated [17,25,26]. Therefore, the current study aimed to identify the clinical significance of NY-ESO-1 and MAGE-A4 expression in STS (UPS, MFS, and MPNST) using pathological specimens from patients treated in our department.

## 2. Results

### 2.1. Rates of Cells Positive for NY-ESO-1 and MAGE-A4

Representative histopathological findings (hematoxylin and eosin [H&E] staining) of UPS (Figure 1a), MFS (Figure 1b), and MPNST (Figure 1c) specimens are shown. Representative images for the positive staining of NY-ESO, MAGE-A4, and Ki67 in each type of STS are shown in Figure 1d–l. Positive staining for NY-ESO and MAGE-A4 was observed in the cytoplasm, whereas that for Ki67 appeared in the nucleus. A summary of the cells positive for NY-ESO, MAGE-A4, and Ki67 expression is shown in Table 1. The average rates of cells positive for NY-ESO, MAGE-A4, and Ki67 expression in all STS types were 18.2%, 39.4%, and 16.8%, respectively. The average rates of cells positive for NY-ESO, MAGE-A4, and Ki67 in the UPS were 9.9%, 34.2%, and 18.8%, respectively. The average rates of cells positive for NY-ESO, MAGE-A4, and Ki67 in MFS were 23.3%, 37.8%, and 15.5%, respectively. The average rates of cells positive for NY-ESO, MAGE-A4, and Ki67 in the MPNST were 30.4%, 61.9%, and 15.9%, respectively.

### 2.2. Immunohistochemical Staining for Each Marker

The rates of all cases positive for NY-ESO, MAGE-A4, and Ki67 were 11/22 (50%), 14/22 (63.6%), and 20/22 (90.9%), respectively. The rates of cases positive for NY-ESO among UPS, MFS, and MPNST were 5/10 (50.0%), 5/9 (55.5%), and 1/3 (33.3%), respectively. The rates of cases positive for MAGE-A4 among the UPS, MFS, and MPNST were 6/10 (60.0%), 6/9 (66.6%), and 2/3 (66.6%), respectively. The rates of cases positive for Ki67 among the UPS, MFS, and MPNST were 10/10 (100%), 7/9 (66.6%), and 3/3 (100%), respectively.

### 2.3. Correlation between the Expression Rates of Two CTAs

A weak correlation was observed between NY-ESO-1 and MAGE-A4 expression levels. (r = 0.22, *p* = 0.14 (Pearson’s test), *p* = 0.001 (Spearman’s test), *p* = 0.10 (Multiple regression test), Figure 2a). The correlation between NY-ESO-1 and Ki67 expression levels was moderate (r = 0.65, *p* = 0.08 (Pearson’s test), *p* = 0.059 (Spearman’s test), *p* = 0.29 (Multiple regression test), Figure 2b). Similarly, the correlation between MAGE-A4 and Ki67 expression levels was also moderate (r = 0.54; *p* = 0.08 (Pearson’s test), *p* = 0.09 (Spearman’s test), *p* = 0.29 (Multiple regression test), Figure 2c).

### 2.4. Correlation between the Expression of NY-ESO-1 or MAGE-A4 and Age

A moderate positive correlation between the rate of cells positive for NY-ESO-1 and age was observed (r = 0.48, *p* = 0.038 (Pearson’s test), *p* = 0.082 (Spearman’s test), *p* = 0.08 (Multiple regression test), Figure 3a), similar to that between the rate of cells positive for MAGE-A4 and age which was strong (r = 0.70, *p* = 0.14 (Pearson’s test), *p* = 0.86 (Spearman’s test), *p* = 0.39 (Multiple regression test), Figure 3b).

### 2.5. Correlation between the Expression of NY-ESO-1 or MAGE-A4 and Tumor Size

The weakly positive correlation between the rate of cells positive for NY-ESO-1 and tumor size was moderate (r = 0.46, *p* = 0.28 (Pearson’s test), *p* = 0.51 (Spearman’s test), *p* = 0.17 (Multiple regression test), Figure 3c) that between the rate of cells positive for MAGE-A4 and tumor size was also moderate (r = 0.61, *p* = 0.61 (Pearson’s test), *p* = 0.65 (Spearman’s test), *p* = 0.27 (Multiple regression test), Figure 3d).

### 2.6. Correlation between the Expression of NY-ESO-1 or MAGE-A4 and Histological Grade

The positive correlation between the rate of cells positive for NY-ESO-1 and histological grade was moderate (r = 0.53, *p* = 0.84 (Pearson’s test), *p* = 0.46 (Spearman’s test), *p* = 0.26 (Multiple regression test), Figure 4a), whereas that between MAGE-A4 expression and histological grade was strong (r = 0.72, *p* = 0.54 (Pearson’s test), *p* = 0.94 (Spearman’s test), *p* = 0.61 (Multiple regression test), Figure 4b).

### 2.7. Correlation between the Expression of NY-ESO-1 or MAGE-A4 and Maximum Standardized Uptake Value

A weak correlation was observed between the rate of cells positive for NY-ESO-1 and maximum standardized uptake value (SUVmax) value (r = 0.26, *p* = 0.83 (Pearson’s test), *p* = 0.01 (Spearman’s test), *p* = 0.84 (Multiple regression test), Figure 4c), whereas there was a moderate positive correlation between the MAGE-A4 expression and SUVmax (r = 0.53, *p* = 0.91 (Pearson’s test), *p* = 0.03 (Spearman’s test), *p* = 0.38 (Multiple regression test), Figure 4d).

### 2.8. Expression of NY-ESO-1 or MAGE-A1 in Continuously Disease-Free Cases or Those with No Evidence of Disease in Comparison with Alive with Disease or Died of Other Cause Cases

The rates of cells positive for NY-ESO-1 in continuously disease free (CDF) or no evidence of disease (NED) cases and in alive with disease (AWD) or died of other cause (DOD) cases were 23.3 ± 31.5 and 0.69 ± 1.35, (average ± standard deviation (SD)), respectively, and there was no significant difference between the two groups (*p* = 0.13, Table 2). The rates of cells positive for MAGE-A4 in CDF or NED cases and in AWD or DOD cases were 34.7 ± 34.1 and 55.7 ± 43.0, (mean ± SD), respectively, and showed no significant difference (*p* = 0.26).

## 3. Discussion

The present study clarified the clinical correlation of NY-ESO-1 and MAGE-A4 expression in the UPS, MFS, and MPNST. High expression of NY-ESO-1 and MAGE-A4 was observed in all types of STSs (UPS, MFS, and MPNST), and a correlation with clinical markers was also observed. The strengths of this study are that we also evaluated the SUVmax value as a clinical item and that Ki67 was used for immunostaining.

NY-ESO-1 and MAGE-A4 have been found to be expressed in synovial sarcoma (49–88%) and myxoid/round cell liposarcoma (35–100%) and suggested to be involved in the pathogenesis of these diseases [16,18,19,25,27]. Kakimoto et al. also reported that both NY-ESO-1 or MAGE-A4 were present in synovial sarcoma and that those CTSs expression were useful in diagnosing of synovial sarcoma [18].

Previous studies reported, an NY-ESO-1 expression rate of 0% for UPS, 0% for MFS, and 8.2% for MPNST [26]. Positive rates of MAGE-A4 of 18.2% in UPS, 0% in MFS, 12–15% in MPNST, and 24.6% in MPNST have also been reported [25,26]. In the present study, we found a high rate of NY-ESO-1 and MAGE-A4 expression ranging from 33.3–50%. The positivity rate of the cells for these CTAs was also higher than that reported previously. These findings suggest that NY-ESO-1 and MAGE-A4 are closely related to the pathogenesis of UPS, MFS, and MPNST and may be useful for diagnosing of these conditions.

In general, older age, larger size, higher histological grade, and higher SUVmax values have been reported as poor prognostic factors for STS [28,29,30,31]. Previous studies have reported that age, tumor size, histological grade, and SUVmax value are dependent adverse prognostic factors in UPS [22,32,33], MFS [33,34,35], and MPNST [33,36,37,38]. In the current study, age, tumor size, histological grade, and SUVmax were correlated with NY-ESO-1 and MAGE-A4 expression. These findings suggest that NY-ESO-1 and MAGE-A4 expression may indicate poor prognosis. If the expression of NY-ESO-1 and MAGE-A4 is indicative of a poor prognosis, then the expression of these CTAs may be higher in remission cases than in non-remission cases. In the present study, no significant difference in the percentage of NY-ESO-1 or MAGE-A4-positive cells was observed between remission and non-remission patients. We believe that this was due to the small sample size and the short follow-up duration in this study.

Interestingly, previous studies have shown that in high-grade sarcoma, patients who were NY-ESO-1 positive or MAGE-A4 positive had a significantly better overall survival rate than those who were negative for these two CTAs [18]. The authors attributed these results to the fact that the NY-ESO-1 and MAGE-A4 positive groups included many patients with synovial sarcoma and myxoid liposarcoma with relatively good prognoses [18]. They also explained why the expression of NY-ESO-1 induced the activation of T cell activity, leading to an anti-tumor response. In the current study, we obtained contrary results compared to those in a previous study [18]. We hypothesized that NY-ESO-1 and MAGE-A4 might function differently in the tumor microenvironment (TME) in highly aggressive sarcomas such as UPS, MFS, and MPNST tumors compared with mild progressive STSs. In addition, T-cell dysfunction, such as functional memory or exhaustion, tolerance, anergy, and senescence [39], might occur depending on the TME of naive T cells that encounter the antigen in highly aggressive STS’. The roles of NY-ESO-1 and MAGE-A4 in various sarcomas should be investigated in the future.

### Limitations

This study had some limitations that need to be addressed. First, the small sample size may have affected the significance of the findings and the power of the statistical analysis in this study. Second, data analysis was performed only using correlation. Moreover, Pearson’s correlation coefficient is not always the most appropriate measure of association, depending on the variable type. Herein, the correlations shown were weak or moderate, and the results of the correlations only between age and NY-ESO-1 were accompanied by those of significance tests. Spearman’s test was also performed, but significant differences only between SUVmax and NY-ESO-1 or MAGE-A4 and between age and NY-ESO-1 were confirmed. Multiple regression test was also performed; however, no significant differences were observed. Third, this was a retrospective study; therefore, there may have been a selection bias when enrolling patients. Fourth, the expression of NY-ESO-1 and MAGE-A4 at the genetic level was not confirmed. Despite these limitations, the current study provided evidence for the involvement of NY-ESO-1 and MAGE-A4 in the TME in highly aggressive STS. However, further studies with multivariate analyses, larger sample sizes, and long-term clinical follow-up durations are warranted. Specifically, significant expression of NY-ESO-1 and MAGE-A4 at the gene level and significant correlation between those expressions and clinical prognostic factors should be confirmed in these highly aggressive STS’. Future randomized controlled trials investigating NY-ESO-1 and MAGE-A4 antibodies are desirable. Furthermore, a clinical trial in addition to conventional chemotherapy could lead to the development of these drugs as therapeutic agents.

## 4. Materials and Methods

Patient characteristics are presented in Table 3. Twenty-two patients with STS were included in this study. All patients were treated at our hospital between January 2006 and December 2019. This study was approved by the Kindai University Ethics Committee (approval number: R03-021; approved on 27 April 2021). Written informed consent was obtained from all patients. Comprehensive consent was obtained from patients who could not sign the consent form.

The median age was 72.5 years (range: 34–101). There were 13 men and nine women. The tumor was located in the upper extremities in four cases, the lower extremities in 13 cases, and the trunk in five cases. The histological grades [40] were grade 1, grade 2, and grade 3 in two, eight, and 12 patients, respectively.

The median tumor diameter was 5.8 cm (range: 1.5–15.1). The median SUVmax value was 7.38 (range: 3.13–24.1). The treatment consisted of wide resection with a flap in two cases, wide resection and skin graft in two cases, additional wide resection with flap in one case, wide resection and postoperative radiotherapy in three cases, wide resection in 11 cases, marginal resection in two cases, and postoperative radiotherapy only in one case. There were eight cases of recurrence and 14 without recurrence. Five patients had metastasis and 17 did not have metastasis.

The final clinical outcomes were CDF in 11 patients, NED in six patients, AWD in three patients, and DOD in two patients.

### 4.1. Immunohistochemical Staining

Immunostaining for NY-ESO-1, MAGE-A4, and Ki67 was performed on pathological specimens harvested at the time of biopsy from patients with STS’, UPS (*n* = 10), MFS (*n* = 9), and MPNST (*n* = 3) treated at our institution.

Tissue sections were formalin-fixed and paraffin-embedded (FFPE). Sections of 4-μm thickness were cut and mounted onto slides. Tissues were deparaffinized, rehydrated, and subjected to endogenous peroxidase inhibition using 3% hydrogen peroxide. Antigen activation was performed using antigen-specific heat treatment at pH 9 as follows: NY-ESO-1 100 °C for 64 min; MAGE-A4, 95 °C for 36 min, p53 95 °C for 64 min, and Ki67 95 °C for 64 min. Following heat activation, the tissue sections were incubated with the following primary antibodies: NY-ESO-1 antibody (mouse monoclonal, E978; Santa Cruz Biotechnology, Santa Cruz, CA, USA) for 32 min at 37 °C, MAGE-A4 antibody (rabbit monoclonal, ab229011; Abcam, Cambridge, UK) for 16 min at 37 °C, and Ki67 antibody (mouse monoclonal, M7240; Agilent Technology, California, USA) for 16 min at 37 °C. The reaction was visualized using 3.3-diaminobenzidine (DAB Substrate Chromogen System; DAKO, Kyoto, Japan), and the sections were counterstained with hematoxylin. Testes were used as positive controls for NY-ESO-1 and MAGE-A4 expression, and tonsils were used as positive controls for Ki67 expression. In all immunohistochemical staining experiments, negative controls were performed using IgG adapted for each stain to check for nonspecific binding. The slides were observed under a microscope (BIOREVO BZ-9000; KEYENCE, Osaka, Japan), and brown granules in the cytoplasm or nuclei indicated a positive staining. Immune marker staining within the tumor was quantified in four representative high-power fields (40× magnification) [41]. The positivity rate of each immune component was calculated. The positivity rate was defined as the number of positive cells/total cell number and was quantified using BIOREVO-BZ 9000 software (Keyence, Osaka City, Japan).

### 4.2. Immunohistochemical Positive Cell Rate

The rate of cells positive for each marker was calculated by securing four representative fields of view for each slide. The rate of positive cells was calculated as the number of positive cells/total cells.

### 4.3. Immunohistochemical Positive Case Rate of Each Marker

Furthermore, a positive rate of ≥5% was considered as a positive case, and the percentage of positive cases for each immunohistochemical marker was calculated.

Correlations of the expression CTAs with various clinicopathological factors including age, tumor size, histological grade, and remission or not in remission were examined.

### 4.4. Statical Analysis

The positivity rate of each marker was plotted, and a correlation diagram was drawn [41]. The coefficient of determination (r) was calculated by drawing an approximation line to examine the correlation between the markers. Pearson’s method of testing for correlations was used to perform tests. The correlation between clinical parameters and the positivity rate of each molecule was also investigated in a similar manner. The r-value criteria were as follows: strong, values between 0.7 and 1.0 (−0.7 and −1.0); moderate, values between 0.3 and 0.7 (0.3 and −0.7); weak, values between 0 and 0.3 (0 and −0.3); no linear relationship, value 0 as previously described [42]. The value of r was calculated by excluding outliers. In addition, Spearman’s rank correlation coefficient test was performed. Multiple regression test was also performed. The chi-square test was used to compare cases in remission with non-remission cases. Statistical significance was set at *p* < 0.05. Analyses were performed using Stat Mate 5.05 (ATMS, Tokyo, Japan).

## 5. Conclusions

We investigated the clinicopathological role of NY-ESO-1 and MAGE-A4 in highly aggressive STS’ (UPS, MFS, and MPNST). NY-ESO-1 and MAGE-A4 may be useful for the diagnosis and prognostication of UPS, MFS, and MPNST.

## Figures and Tables

**Figure 1 diagnostics-12-00733-f001:**
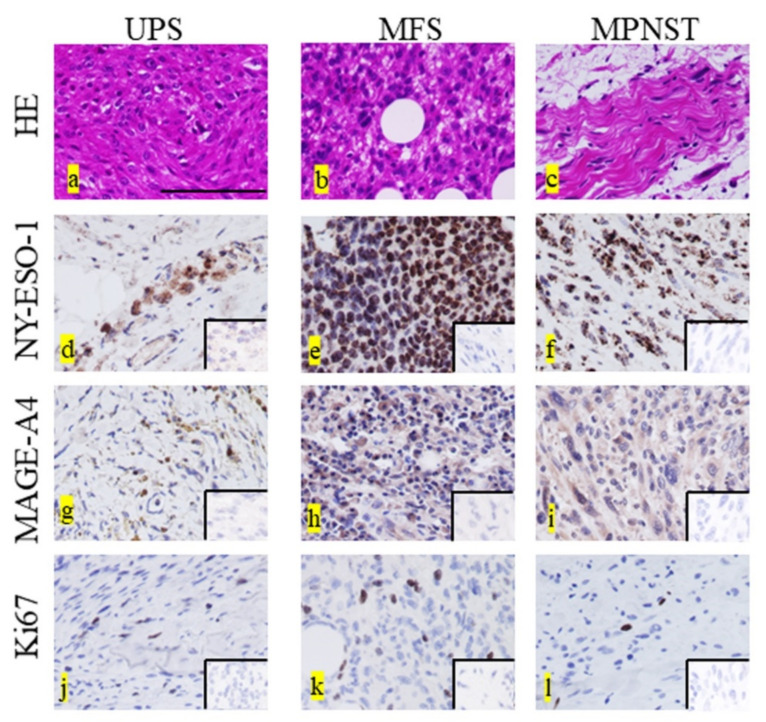
Representative histopathological findings in undifferentiated pleomorphic sarcoma (UPS) (**a**), myxofibrosarcoma (MFS) (**b**), and malignant peripheral sheath tumor (MPNST) (**c**). Hematoxylin and eosin (HE) staining (**a**–**c**). Representative NY-ESO-1-positive histological findings in UPS (**d**), MFS (**e**), and MPNST (**f**). Representative MAGE-A4-positive histological findings in UPS (**g**), MFS (**h**), and MPNST (**i**). Representative Ki67-positive histological findings in UPS (**j**), MFS (**k**), and MPNST (**l**). NY-ESO and MAGE-A4 show staining in the cytoplasm, whereas Ki67 shows staining in the nucleus (**d**–**l**). The lower right inset image was used as the negative control for each immunostaining image (**d**–**l**). Scale bar = 100 µm.

**Figure 2 diagnostics-12-00733-f002:**
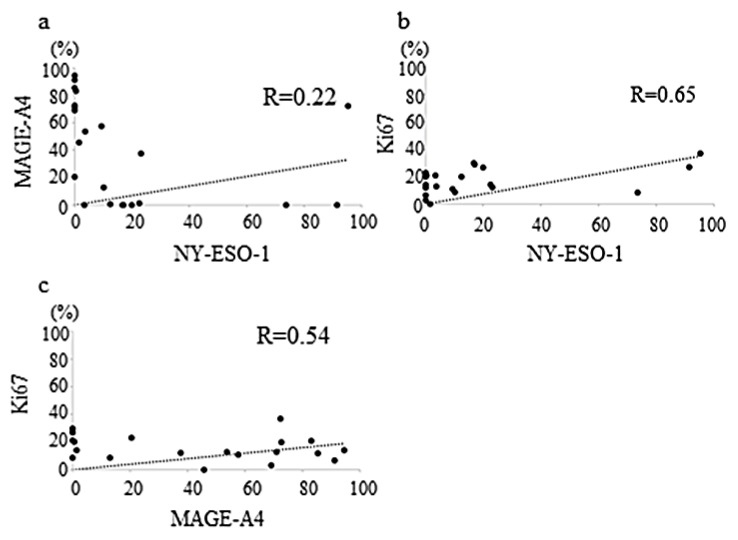
Graphs showing no positive correlation between the NY-ESO-1 and MAGE-A4 expression levels (r = 0.22) in sarcomas (**a**); weak positive correlation between the NY-ESO-1 and Ki67 expression levels (r = 0.65) in sarcomas (**b**); and weak positive correlation between the MAGE-A4 and Ki67 expression levels (r = 0.54) in sarcomas (**c**).

**Figure 3 diagnostics-12-00733-f003:**
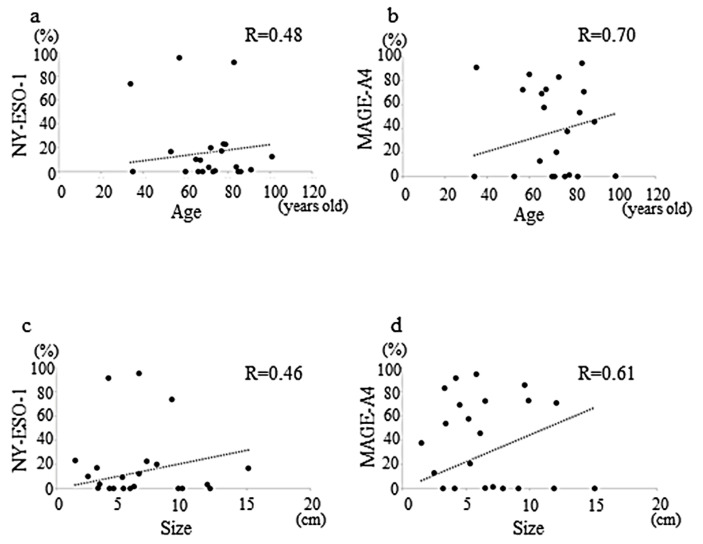
Graphs showing the weakly positive correlation between age and NY-ESO-1 expression (r = 0.48) in sarcomas (**a**); weak positive correlation between the age and MAGE-A4 expression (r = 0.70) in sarcomas (**b**); weak positive correlation between the tumor size and NY-ESO-1 expression (r = 0.46) in sarcomas (**c**); weak positive correlation between the tumor size and MAGE-A4 expression (r = 0.61) in sarcomas (**d**).

**Figure 4 diagnostics-12-00733-f004:**
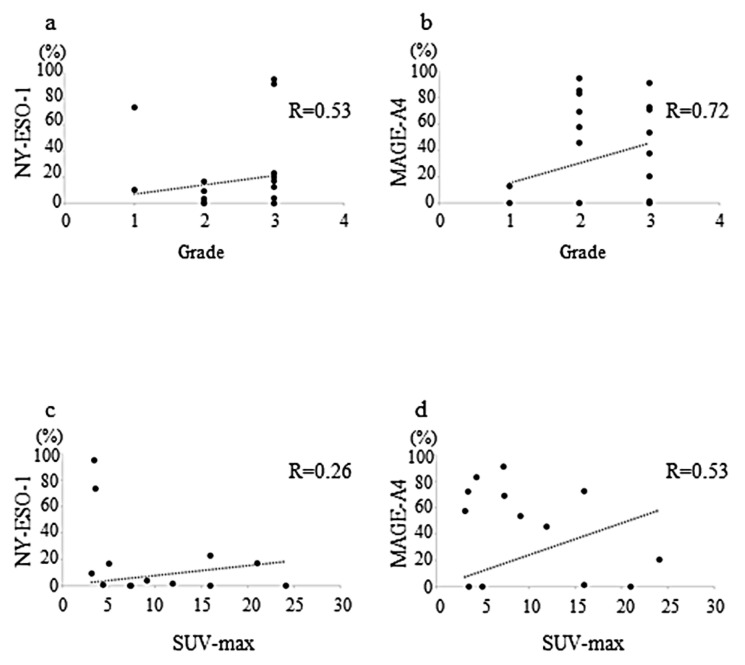
Graphs showing a weakly positive correlation between histological grade and NY-ESO-1 expression (r = 0.53) in sarcomas (**a**); very strong positive correlation between the histological grade and MAGE-A4 expression (r = 0.72) in sarcomas (**b**); no correlation was observed between the maximum standardized uptake value (SUVmax) and NY-ESO-1 expression (r = 0.26) in sarcomas (**c**); weak positive correlation between the SUVmax and MAGE-A4 expression (r = 0.53) in sarcomas (**d**).

**Table 1 diagnostics-12-00733-t001:** Summary of positive cell rates.

Positive Cell Rate (%)	
NY-ESO-1	
0–10	12
10–50	7
≥50	3
MAGE-A4	
0–10	8
10–50	4
≥50	10
Ki67	
0–10	5
10–50	17
≥50	0

NY-ESO-1; New York esophageal squamous cell carcinoma-1, MAGE-A4; melanoma-associated antigen A4.

**Table 2 diagnostics-12-00733-t002:** Expression of NY-ESO-1 or MAGE-A1 in continuously disease-free cases or those without evidence of disease in comparison with alive with disease or died of other cause cases.

Molecules	NY-ESO-1		MAGE-A4	
Outcome	CDF or NED	AWD or DOD	CDF or NED	AWD or DOD
Average (%)	23.3	0.69	34.7	55.7
S.D.	31.5	1.35	34.1	43.0
*p*-value	0.13		0.26	

S.D, standard deviation; NY-ESO-1, New York esophageal squamous cell carcinoma-1; MAGE-A4, melanoma antigen gene A4; CDF, continuous disease free; NED, no evidence of disease; AWD, alive with disease; DOD, dead of disease.

**Table 3 diagnostics-12-00733-t003:** Characteristics of the study population.

Factor	Patients, *n*
Age (years)	
>70	9
≤70	13
Sex	
Male	13
Female	9
Tumor site	
Arms	4
Legs	9
Trunk	9
Histological type	
MFS	9
UPS	10
MPNST	3
Histological grade	
Grade 1	2
Grade 2	8
Grade 3	12
Tumor size (cm)	
<5	8
5–10	11
>10	3
SUV-max	
<5	4
5–10	4
>10	5
Treatment	
Wide resection, Flap	4
Wide resection, Post operative radition	3
Wide resection	10
Wide resection, skin graft	2
Marginal resection	2
Radiation	1
Recurrence	
(+)	8
(−)	14
Metastasis	
(+)	5
(−)	17
Outcome	
CDF	11
NED	6
AWD	3
DOD	2
Follow-up period (years)	
<3	8
≥3	14

MFS; myxofibrosarcoma, UPS; undifferentiated pleomorphic sarcoma, MPNST; malignant peripheral nerve sheath tumor, SUV-max; Maximum standardized uptake value -max, Op; operation, CDF; continuous disease free, NED; no evidence of disease, AWD; alive with disease, DOD; dead of disease.

## Data Availability

The datasets used and/or analyzed during the current study are available from the corresponding author on reasonable request.

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
