# Peer review of "Clinicopathological Assessment of Cancer/Testis Antigens NY-ESO-1 and MAGE-A4 in Highly Aggressive Soft Tissue Sarcomas"

_diagnostics, 2022, doi:10.3390/diagnostics12030733_

Round 1

Reviewer 1 Report

In this article, the author tried to evaluate correlation between the two cancer/testis antigens, NY-ESO-1 and MAGE-A4 in soft tissue scarcoma. Consequently, the manuscript is not ready for publication. I try to list some elements to sustain the fact that it is mature neither on the form nor on the content.

Major Points:

  1. Frist, the article is summited to “Diagnostics”, but the format of the article which the authors uploaded is “International Journal of Molecular Science”.
  2. Although the authors try to evaluate the possible function of NY-ESO-1 and MAGE-A4 in soft tissue sarcoma. The correlation between NY-ESO-1 and MAGE-A4 is weak positive correlation. Besides, the similar results were published in other journal (ONCOLOGY LETTERS 17: 3937-3943, 2019) and the sample sizes are much more than the present article.
  3. The result of “2.8. Expression of NY-ESO-1 or MAGE-A1 in continuously disease free or no evidence of disease cases in comparison with alive with disease or died of other cause cases” did not present with tables or figures. Please add these information.
  4. The author did not discuss the result of 2.8 in the discussion. Please add it to discussion.
  5. As the authors mentioned, the small sample size may have affected the significance of the findings and the power of the statistical analysis in this study. Please increase the sample size to further confirm the results.

Author Response

Thank you very much for reviewing our manuscript. We would like to thank you for the peer review. Your helpful and insightful comments have helped to greatly improve the quality of the manuscript. Deletions are in the manuscript are indicated using strikethrough text and red font, while additions are indicated using blue font. Kindly consider reviewing our revised manuscript.

#Reviewer1

In this article, the author tried to evaluate correlation between the two cancer/testis antigens, NY-ESO-1 and MAGE-A4 in soft tissue sarcoma. Consequently, the manuscript is not ready for publication. I try to list some elements to sustain the fact that it is mature neither on the form nor on the content.

Major Points:

  1. Frist, the article is summited to “Diagnostics”, but the format of the article which the authors uploaded is “International Journal of Molecular Science”.

Authors’ response and action:

Thank you so much for pointing this out. We have ensured that the revised manuscript is formatted per the template and formatting guidelines provided by the journal “Diagnostics.”

  1. Although the authors try to evaluate the possible function of NY-ESO-1 and MAGE-A4 in soft tissue sarcoma. The correlation between NY-ESO-1 and MAGE-A4 is weak positive correlation. Besides, the similar results were published in other journal (ONCOLOGY LETTERS 17: 3937-3943, 2019) and the sample sizes are much more than the present article.

Authors’ response and action:

Thank you for your valuable comment. As you pointed out, the correlation between NY-ESO-1 and MAGE-A4 was weakly positive. Further, because the other reviewer recommended it, we revised the r-value criteria. Accordingly, no correlation was observed between NY-ESO-1 and MAGE-A4. Additionally, the sample size is relatively small. However, unlike the study by Kakimoto et al. (reference #18), our study describes the results of immunostaining for Ki67, and we also evaluated the relationship between SUVmax value and clinical characteristics. These are the strengths of this study. We have added this information to the beginning of the discussion section. In addition, the positivity rate in this study was higher than that previously reported. We believe that this indicates a strong involvement of NY-ESO and MAGE-A4 in STSs. The small cohort is a limitation of this study; we have modified the text per the aforementioned description as follows:

Discussion section (page 8, lines 169–171; page 8, lines 179–183)

“The strengths of this study are that we also evaluated the SUVmax value as a clinical item and that Ki67 was used for immunostaining.”

“In the present study, we found a high rate of NY-ESO-1 and MAGE-A4 expression, ranging from 33.3-50%. The positivity rate of the cells for these CTAs was also higher than that reported previously. These findings suggest that NY-ESO-1 and MAGE-A4 are closely related to the pathogenesis of UPS, MFS, and MPNST and may be useful for the diagnosis of these conditions.”

Limitations (page 9, lines 212–­224)

“This study had some limitations that need to be addressed. First, the small sample size may have affected the significance of the findings and the power of the statistical analysis in this study. Second, Pearson’s correlation coefficient is not always the most appropriate measure of association, depending on the type of variable. Herein, the correlations shown were mainly weak, ereinHereinand the results of the correlations were not accompanied by significance tests. Third, this was a retrospective study; therefore, there may have been a selection bias when enrolling patients. Fourth, the expression of NY-ESO-1 and MAGE-A4 at the genetic level was not confirmed. Despite these limitations, the current study provided evidence for the involvement of NY-ESO-1 and MAGE-A4 in the tumor microenvironment in highly aggressive STS. However, further studies with multivariate analyses, larger sample sizes, and long-term clinical follow-up durations are warranted.”

  1. The result of “2.8. Expression of NY-ESO-1 or MAGE-A1 in continuously disease free or no evidence of disease cases in comparison with alive with disease or died of other cause cases” did not present with tables or figures. Please add these information.

Authors’ response and action:

Thank you for your important suggestion. We have added a new table, “Table 3,” in the revised manuscript, as follows (page 8):

              Table 3. Expression of NY-ESO-1 or MAGE-A1 in continuously disease-free cases or               those without evidence of disease in comparison with alive with disease or died of other      cause cases

Molecules

NY-ESO-1

MAGE-A4

Outcome

CDF or NED

AWD or DOD

CDF or NED

AWD or DOD

Average (%)

23.3

0.69

34.7

55.7

S.D.

31.5

1.35

34.1

43.0

p-value

0.13

0.26

              S.D; standard deviation, NY-ESO-1; New York esophageal squamous cell carcinoma-1,           MAGE-A4; melanoma antigen gene A4, CDF; continuously disease free, NED; no               evidence of disease, AWD; alive with disease, DOD; dead of disease

  1. The author did not discuss the result of 2.8 in the discussion. Please add it to discussion.

Authors’ response and action:

Thank you for pointing this out. In the current study, we compared the positive rates of NY-ESO and MAGE-A4 in CDF and NED cases with AWD and DOD cases; no significant difference was observed in these rates. A comparative study with a larger sample size and a longer follow-up duration is warranted. We have added this information to the text as follows:

Discussion (page 8, lines 190–194)

“If the expression of NY-ESO-1 and MAGE-A4 is indicative of a poor prognosis, the expression of these CTAs may be higher in remission cases than in non-remission cases. In the present study, no significant difference in the percentage of NY-ESO-1 or MAGE-A4-positive cells was observed between remission and non-remission patients. We believe that this was due to the small sample size and the short follow-up duration in this study.”

Limitations (page 9, lines 212–­224)

“This study had some limitations that need to be addressed. First, the small sample size may have affected the significance of the findings and the power of the statistical analysis in this study. Second, Pearson’s correlation coefficient is not always the most appropriate measure of association, depending on the type of variable. Herein, the correlations shown were mainly weak, ereinHereinand the results of the correlations were not accompanied by significance tests. Third, this was a retrospective study; therefore, there may have been a selection bias when enrolling patients. Fourth, the expression of NY-ESO-1 and MAGE-A4 at the genetic level was not confirmed. Despite these limitations, the current study provided evidence for the involvement of NY-ESO-1 and MAGE-A4 in the tumor microenvironment in highly aggressive STS. However, further studies with multivariate analyses, larger sample sizes, and long-term clinical follow-up durations are warranted.”

  1. As the authors mentioned, the small sample size may have affected the significance of the findings and the power of the statistical analysis in this study. Please increase the sample size to further confirm the results.

Authors’ response and action:

Thank you so much for your suggestion. As you have pointed out, the sample size is small and presents a statistical problem. We wish to address this issue in our future studies. This has been added as a limitation of the study, and we weakened the overall argument and conclusions of the study.

Limitations (page 9, lines 212–­224)

“This study had some limitations that need to be addressed. First, the small sample size may have affected the significance of the findings and the power of the statistical analysis in this study. Second, Pearson’s correlation coefficient is not always the most appropriate measure of association, depending on the type of variable. Herein, the correlations shown were mainly weak, ereinHereinand the results of the correlations were not accompanied by significance tests. Third, this was a retrospective study; therefore, there may have been a selection bias when enrolling patients. Fourth, the expression of NY-ESO-1 and MAGE-A4 at the genetic level was not confirmed. Despite these limitations, the current study provided evidence for the involvement of NY-ESO-1 and MAGE-A4 in the tumor microenvironment in highly aggressive STS. However, further studies with multivariate analyses, larger sample sizes, and long-term clinical follow-up durations are warranted.”

Conclusions (page 10, lines 294–296)

“We investigated the clinicopathological role of NY-ESO-1 and MAGE-A4 in highly aggressive STSs (UPS, MFS, and MPNST). NY-ESO-1 and MAGE-A4 may be useful for the diagnosis and prognostication of UPS, MFS, and MPNST.”

Reviewer 2 Report

In this article, the authors investigate the association between the expression of three cancer/testis antigens (CTA) and soft tissue sarcoma (STS). The established correlations with various strengths between the immunostaining with the markers and multiple types of STS’s patient tissue. Moreover, there were some associations between the presence of the antigen and histological/clinical characteristics of the patients.

The findings are well presented and of value; however, the article's conclusion is too strong for the data presented.

  • I am referring to the final sentence of the abstract, “Our results suggest that NY-ESO-1 and MAGE-A4 expression is involved in the pathogenesis 23 of UPS, NFS, and MPNST.” I think this is not supported by the data presented in the article. The authors should conclude a causal link or specify “pathogenesis” as a role of the antigens based on the data they give alone.

In addition, I would like to raise a few issues and suggestions on the use of statistical associations and tests.

  • Correlation analysis is not sufficient and can be improved upon (I suggest using multivariate analysis)
  • A significance test of the associations was not performed, or at least not reported in some parts of the results section.
  • Pearson’s correlation coefficient is not always the most appropriate measure of association, depending on the type of variable.
  • Determining the significance of the “r” value is arbitrary. I suggest using more formal criteria to establish the strong, weak, or none categorization.

Author Response

Thank you very much for reviewing our manuscript. We would like to thank you for the peer review. Your helpful and insightful comments have helped to greatly improve the quality of the manuscript. Deletions are in the manuscript are indicated using strikethrough text and red font, while additions are indicated using blue font. Kindly consider reviewing our revised manuscript.

In this article, the authors investigate the association between the expression of three cancer/testis antigens (CTA) and soft tissue sarcoma (STS). The established correlations with various strengths between the immunostaining with the markers and multiple types of STS’s patient tissue. Moreover, there were some associations between the presence of the antigen and histological/clinical characteristics of the patients.

The findings are well presented and of value; however, the article's conclusion is too strong for the data presented.

  • I am referring to the final sentence of the abstract, “Our results suggest that NY-ESO-1 and MAGE-A4 expression is involved in the pathogenesis 23 of UPS, NFS, and MPNST.” I think this is not supported by the data presented in the article. The authors should conclude a causal link or specify “pathogenesis” as a role of the antigens based on the data they give alone.

Authors’ response and action:

Thank you so much for your useful comment. As you pointed out, the last sentence of the original abstract, “Our results suggest that NY-ESO-1 and MAGE-A4 expression is involved in the pathogenesis of UPS, NFS, and MPNST,” was too strong a conclusion based on our results. This statement was revised as follows (page 10, lines 291–294):

“We investigated the clinicopathological role of NY-ESO-1 and MAGE-A4 in highly aggressive STSs (UPS, MFS, and MPNST). NY-ESO-1 and MAGE-A4 may be useful for the diagnosis and prognostication of UPS, MFS, and MPNST.”

  • In addition, I would like to raise a few issues and suggestions on the use of statistical associations and tests.
  • Correlation analysis is not sufficient and can be improved upon (I suggest using multivariate analysis)

Authors’ response and action:

Thank you very much for your recommendations. As you pointed out, the correlation analysis alone is not sufficient; we wish to perform a multivariate analysis in our future research, using a larger sample size. We modified the limitations section as follows:

Limitations (page 9, lines 212–­224)

“This study had some limitations that need to be addressed. First, the small sample size may have affected the significance of the findings and the power of the statistical analysis in this study. Second, Pearson’s correlation coefficient is not always the most appropriate measure of association, depending on the type of variable. Herein, the correlations shown were mainly weak, ereinHereinand the results of the correlations were not accompanied by significance tests. Third, this was a retrospective study; therefore, there may have been a selection bias when enrolling patients. Fourth, the expression of NY-ESO-1 and MAGE-A4 at the genetic level was not confirmed. Despite these limitations, the current study provided evidence for the involvement of NY-ESO-1 and MAGE-A4 in the tumor microenvironment in highly aggressive STS. However, further studies with multivariate analyses, larger sample sizes, and long-term clinical follow-up durations are warranted.”

  • A significance test of the associations was not performed, or at least not reported in some parts of the results section.

Authors’ response and action:

Thank you very much for your pointing out. As you pointed out, significance tests were not conducted for the associations observed; we have added this as a limitation. In addition, we revised the claim of this paper to be weaker. We wish to conduct further, similar studies with a larger sample size and involving significance tests. The following limitation was added (page 9, lines 215–217):

“Herein, the correlations shown were mainly weak, ereinHereinand the results of the correlations were not accompanied by significance tests.”

  • Pearson’s correlation coefficient is not always the most appropriate measure of association, depending on the type of variable.

Authors’ response and action:

Thank you so much for your comment. As you pointed out, Pearson’s correlation coefficient is not always the most appropriate measure of association, depending on the type of variable. This has been included as a limitation as follows (page 9, lines 214–217):

“Second, Pearson’s correlation coefficient is not always the most appropriate measure of association, depending on the type of variable. Herein, the correlations shown were mainly weak, ereinHereinand the results of the correlations were not accompanied by significance tests.”

  • Determining the significance of the “r” value is arbitrary. I suggest using more formal criteria to establish the strong, weak, or none categorization.

Authors’ response and action:

Thank you very much for your useful comment and suggestion. We re-calculated the r-value by excluding the outliers. This information has been added to the “Materials and Methods” section. We also revised the classification as follows strong, 0.7>; weak 0.7 >weak ≥0.3; no correlation, <0.3. Accordingly, the strength of each correlation was revised in the results section.

Material and Methods (page 10, lines 287–288)

“The value of r was calculated by excluding outliers.”

Round 2

Reviewer 1 Report

No further comments.

Author Response

Thank you very much for reviewing our manuscript once again. We really appreciate you. We have revised the text to make it readable again. 

Reviewer 2 Report

In the revised version of the manuscript, the authors

  • Changed some of the findings and conclusions to make weaker claims
  • Expanded the limitation section to include some of the issues raised in my first review.

The author's response to the review merely pointed to the minor changes to the text to either weaken the claim or add to the limitations. I do not think the authors engaged with the critics or attempted to address them. Most of the points raised earlier hold, in particular

  • The use of correlational analysis only
  • The lack of significant testing
  • Not using formal criteria to judge the magnitude of the correlations

I do not think the revised manuscript warrants a change in the initial judgment of the validity of the manuscript.

Author Response

We sincerely thank you for reviewing our manuscript again. Your insightful comments and suggestions have helped us in greatly improving the quality of the manuscript. Deletions in the revised manuscript are indicated as strikethrough text in red font, whereas changes are indicated using blue font.

Reviewer#2

Comments and Suggestions for Authors

In the revised version of the manuscript, the authors

Changed some of the findings and conclusions to make weaker claims

Expanded the limitation section to include some of the issues raised in my first review.

The author's response to the review merely pointed to the minor changes to the text to either weaken the claim or add to the limitations. I do not think the authors engaged with the critics or attempted to address them. Most of the points raised earlier hold, in particular

The use of correlational analysis only

The lack of significant testing

Not using formal criteria to judge the magnitude of the correlations

I do not think the revised manuscript warrants a change in the initial judgment of the validity of the manuscript.

Authors’ response: As you pointed out, our study has some shortcomings: we performed only correlation analysis, and we did not conduct any significance test. Moreover, we did not use formal criteria to determine the magnitude of the correlation. Therefore, we reconsidered the formal criteria for correlation, and we have cited reference 42. We also conducted Pearson's significance test. In addition, we performed Pearson's significance test, the results of which showed a significant difference in the correlation between age and NY-ESO-1 (p=0.038). Furthermore, we found significant differences in the correlation between SUVmax and NY-ESO-1 and MAGE-A4 (p=0.01, p=0.03). Other than these findings, no significant differences were found and only correlations were used in the data analysis. We have described this information below as a limitation of this study.

Second, data analysis was performed only using correlation. Moreover, Pearson’s correlation coefficient is not always the most appropriate measure of association, depending on the type of variable. Herein, the correlations shown were weak or moderate, and the results of the correlations only between age and NY-ESO-1 were were not accompanied by those of significance tests. Spearman's test was also performed, but significant differences only between SUVmax and NY-ESO-1 or MAGE-A4 and between age and NY-ESO-1 were confirmed.”

Round 3

Reviewer 2 Report

In the revised manuscript, the authors addressed two issues

  • They performed significance tests on the correlations
  • They rephrased the claims of (weak, moderate, or strong) based on a referenced study

In addition, the authors modified the main text accordingly.

One issue from the initial review remains, however. The manuscript doesn't present a uni- or multi-variate analysis. Although the authors acknowledge this shortcoming in the limitations section, they do not justify the lack of attempt. The small sample size, which is also recognized, would not make this sort of analysis any less valid than mere correlations.

Since the author chose to weaken the initial claims, they might supplement the study's conclusion of what sort of future research would support their conclusions. And more importantly, they can be specific about how this particular study can be helpful to in research or application in diagnosing the disease.

Author Response

Thank you very much for reviewing our manuscript once again. Your helpful and insightful comments have helped to greatly improve the quality of the manuscript. Deletions in the manuscript are indicated using strikethrough text and red font, while additions are indicated using the blue font. Kindly consider reviewing our revised manuscript again.

Author’s response:

Thank you very much for pointing this out. As you pointed out, the lack of multivariate analysis is one of the major problems. Hence, we have performed a multivariate analysis accordingly.

The results are shown below.

NY-ESO-1/MAGE-A4: p=0.10

NY-ESO-1/Ki67:p=0.29

MAGE-A4/Ki67: p=0.29

NY-ESO-1/Age: 0.08

NY-ESO-1/Size: 0.17

NY-ESO-1/Grade: 0.26

NY-ESO-1/SUVmax:0.84

MAGE-A4/Age: 0.39

MAGE-A4/Size: 0.27

MAGE-A4/Grade: 0.61

MAGE-A4/SUVmax:0.38”

We did not observe significant differences in all of them. We have added these results in the results part as follows.

“A weak correlation was observed between NY-ESO-1 and MAGE-A4 expression levels. (r = 0.22, p = 0.14 [Pearson’s test], p = 0.001 [Spearman’s test], p=0.10 [Multiple regression test], Figure 2a). The correlation between NY-ESO-1 and Ki67 expression levels was moderate (r = 0.65, p = 0.08 [Pearson’s test], p = 0.059 [Spearman’s test], p=0.29 [Multiple regression test], Figure 2b). Similarly, the correlation between MAGE-A4 and Ki67 expression levels was also moderate (r = 0.54; p = 0.08 [Pearson’s test], p = 0.09 [Spearman’s test], p=0.29 [Multiple regression test], Figure 2c).

A moderate positive correlation between the rate of cells positive for NY-ESO-1 and age was observed (r = 0.48, p = 0.038 [Pearson’s test], p = 0.082 [Spearman’s test], p=0.08 [Multiple regression test], Figure 3a), similar to that between the rate of cells positive for MAGE-A4 and age which was strong (r = 0.70, p = 0.14 [Pearson’s test], p = 0.86 [Spearman’s test], p=0.39 [Multiple regression test], Figure 3b).

The weakly positive correlation between the rate of cells positive for NY-ESO-1 and tumor size was moderate (r = 0.46, p = 0.28 [Pearson’s test], p = 0.51 [Spearman’s test], p=0.17 [Multiple regression test], Figure 3c) that between the rate of cells positive for MAGE-A4 and tumor size was also moderate (r = 0.61, p = 0.61 [Pearson’s test], p = 0.65 [Spearman’s test], p=0.27 [Multiple regression test], Figure 3d).

The positive correlation between the rate of cells positive for NY-ESO-1 and histological grade was moderate (r = 0.53, p = 0.84 [Pearson’s test], p = 0.46 [Spearman’s test], p=0.26 [Multiple regression test], Figure 4a), whereas that between MAGE-A4 expression and histological grade was strong (r = 0.72, p = 0.54 [Pearson’s test], p = 0.94 [Spearman’s test], p=0.61 [Multiple regression test], Figure 4b). A weak correlation was observed between the rate of cells positive for NY-ESO-1 and maximum standardized uptake value (SUVmax) value (r = 0.26, p = 0.83 [Pearson’s test], p = 0.01 [Spearman’s test], p=0.84 [Multiple regression test], Figure 4c), whereas there was a moderate positive correlation between the MAGE-A4 expression and SUVmax (r =0.53, p = 0.91 [Pearson’s test], p = 0.03 [Spearman’s test], p=0.38 [Multiple regression test], Figure 4d).

We have added this sentence to the limitation as follows.

“Multiple regression test was also performed; however, no significant differences were observed.” (Lines 228-229)

In addition, it will be necessary to investigate the expression of NY-ESO-1 in STSs and MAGE-A4 at the gene level and demonstrate the correlation between these expressions. The results should be used to conduct a randomized controlled trial using anti-NY-ESO-1 and anti-MAGE-A4 drugs, and lead to the development of therapeutic agents.

We have also added these points to the discussion part as follows.

“Specifically, significant expression of NY-ESO-1 and MAGE-A4 at the gene level and significant correlation between those expressions and clinical prognostic factors should be confirmed in these highly aggressive STSs. Future randomized controlled trials investigating NY-ESO-1 and MAGE-A4 antibodies are desirable. Furthermore, a clinical trial in addition to conventional chemotherapy could lead to the development of these drugs as therapeutic agents.” (Lines 234-240)
